

# Evaluating acoustic signals to reduce avian collision risk

Robin G. Thady[1], Lauren C. Emerson[1] and John P. Swaddle[1,2]

[1] Biology Department, William & Mary, Williamsburg, VA, United States of America
[2] Institute for Integrative Conservation, William & Mary, Williamsburg, VA, United States of America

## ABSTRACT

Collisions with human-made structures are responsible for billions of bird deaths each year, resulting in ecological damage as well as regulatory and financial burdens to many industries. Acoustic signals can alert birds to obstacles in their flight paths in order to mitigate collisions, but these signals should be tailored to the sensory ecology of birds in flight as the effectiveness of various acoustic signals potentially depends on the influence of background noise and the relative ability of various sound types to propagate within a landscape. We measured changes in flight behaviors from zebra finches released into a flight corridor containing a physical obstacle, either in no-additional-sound control conditions or when exposed to one of four acoustic signals. We selected signals to test two frequency ranges (4–6 kHz or 6–8 kHz) and two temporal modulation patterns (broadband or frequency-modulated oscillating) to determine whether any particular combination of sound attributes elicited the strongest collision avoidance behaviors. We found that, relative to control flights, all sound treatments caused birds to maintain a greater distance from hazards and to adjust their flight trajectories before coming close to obstacles. There were no statistical differences among different sound treatments, but consistent trends within the data suggest that the 4–6 kHz frequency-modulated oscillating signal elicited the strongest avoidance behaviors. We conclude that a variety of acoustic signals can be effective as avian collision deterrents, at least in the context in which we tested these birds. These results may be most directly applicable in scenarios when birds are at risk of collisions with solid structures, such as wind turbines and communication towers, as opposed to window collisions or collisions involving artificial lighting. We recommend the incorporation of acoustic signals into multimodal collision deterrents and demonstrate the value of using behavioral data to assess collision risk.

Corresponding author
Robin G. Thady,
rgthady@email.wm.edu

## INTRODUCTION

North American bird populations have declined by nearly 30% in the last 50 years (*Rosenberg et al., 2019*), largely resulting from anthropogenic stressors. Collisions with humanmade structures such as wind turbines, power lines, communication towers, aircraft, buildings, and windows are among the most notorious sources of accidental bird mortality (*Manville, 2005*; *Zakrajsek & Bissonette, 2005*; *Klem, 2008*), causing hundreds of millions of bird deaths in the United States annually (*Loss et al., 2014*). The resulting loss of avian biodiversity has ecological and conservation consequences (*Şekercioğlu, Daily*

& Ehrlich, 2004) and damages from collisions themselves impose financial burdens on a variety of industries and raise potential threats to human safety, such as through strikes to aircraft mid-flight (Allan, 2000; Richardson & West, 2000; Thorpe, 2012). It is essential to reduce the incidence of bird collisions in order to prevent loss of crucial ecosystem functions (Van Bael et al., 2008; Garcia, Zamora & Amico, 2010; Kale et al., 2014), appease economic stakeholders (Allan, 2000), and improve the safety and efficiency of commerce and transportation (Thorpe, 2012).

In particular, windowed building and automobile collisions are responsible for the majority of these fatal bird strikes and have been linked to hundreds of millions of deaths annually in the United States—compared with perhaps tens of millions of deaths attributable to all communication tower, power line, and wind turbine collisions taken together (Loss, Will & Marra, 2015). Despite the relatively lower incidence of fatal bird strikes with these taller objects in open landscapes, the associated collision mortality can have adverse effects on local ecosystems and populations (Drewitt & Langston, 2008; Eichhorn et al., 2012). The impacts are often the greatest on birds that are larger in size, longer-lived, and have longer gestation periods (D'Amico et al., 2019), as well as on migratory species and birds associated with agricultural habitats and other human-modified open landscapes, such as those in the order Accipitriformes (Thaxter et al., 2017). These context-specific mortalities can therefore have detectable population-level effects on particularly susceptible species.

Bird strikes will likely continue to occur with increasing frequency as humans continue to expand urban areas and introduce human-built structures into avian habitats. In order to develop effective collision mitigation technology, it is first necessary to understand how birds perceive their surroundings and the risk of collision. Human development has fragmented landscapes with tall objects to which birds are not adapted, and avian visual systems limit birds' ability to detect these obstacles, rendering them vulnerable to collisions.

Birds' visual anatomy and in-flight behaviors together determine the extent to which they can detect physical obstacles and perceive potential threats mid-flight, particularly when passing through relatively open airspace in longer bouts of flight. Although considerable interspecific differences exist in the perceptual capacity of the eyes—with birds of prey notably tending to have the greatest visual acuity (Mitkus et al., 2018)—certain commonalities in skull morphology and direction of gaze exist across groups that limit birds' ability to detect objects in their flight paths. Most birds have eyes oriented laterally on their heads with non-parallel optical systems due to the degree of physical separation between their eyes. As a result, the binocular fields of many birds are narrow (Martin, 2007). Their limited frontal detection is further encumbered because birds must use their peripheral vision to look in the forward direction, which tends to be less accurate than vision from the center of the eye (Martin, 2007). In addition, the narrowness and limited vertical extent of the binocular field create considerable blind spots in the posterior, dorsal, and ventral directions of most birds (Martin, 2014). Many birds frequently look downward or turn their heads to the side while in longer bouts of flight and in open airspace (Martin & Shaw, 2010), and these head movements account for the majority of birds' linear gaze

(*Gioanni, 1988*; *Eckmeier et al., 2008*). Therefore, even subtle adjustments to the position of the head while in flight may render birds relatively blind to the direction of travel.

Historically, frontal detection in open airspace has likely been relatively less important to high-flying or migrating birds due to the lack of tall physical obstructions in natural landscapes. Because of this and the inherent limitations to binocular vision, birds in flight may rely more heavily on their lateral visual fields to detect conspecifics, food, and predators, thus reducing their attention to the forward direction (*Martin, 2011*). Taken together, birds flying through open airspace likely have a limited ability to detect obstacles directly in front of them with sufficient time to react and avoid a collision.

Because birds' visual perception is often insufficient to detect the collision hazards themselves, collision mitigation strategies that are solely intended to engage with the visual system—such as ultraviolet lights, lasers, and boldly-patterned decals—have been met with only limited, context-dependent effectiveness and reveal great interspecific variation in success (*Blackwell, Bernhardt & Dolbeer, 2002*; *Håstad & Ödeen, 2014*; *Martin, 2014*; *Habberfield & St. Clair, 2016*), especially as these visual deterrents are generally placed on the very same objects that the birds already fail to perceive. Additional alarm or diversion strategies can further increase the chance of detection (*Martin, 2011*). Multimodal signals may help resolve the shortcomings of current collision deterrents by engaging with multiple sensory systems at the same time to increase avian attention to the surrounding environment (*Boycott et al., 2021*). In particular, sound could be used as a preliminary signal to birds as they approach tall objects, raising their awareness so that they can visually detect the threat and change direction before a collision can occur (*Swaddle & Ingrassia, 2017*).

According to laboratory studies of birds generally in the absence of background noise, most birds are more sensitive to frequencies between 1 and 5 kHz and can often hear sounds up to 8–10 kHz (*Dooling, 2002*). However, birds likely experience different sound environments when in flight in the wild than under these laboratory conditions. A free flying bird's sound environment likely includes sounds from flapping wings, moving air currents, and lower-frequency ambient noise. Hence, the lower end (1–3 kHz) of the hearing ranges indicated above are likely partially masked for flying birds. As a result, it may be necessary for acoustic signals to be higher in frequency than the documented peak sensitivity range of most birds in order to be more easily detected while birds fly through open landscapes.

In addition to the frequency characteristics of potential signals, the temporal patterns of sounds (hereafter "sound shapes") may impact the degree to which they effectively elicit collision avoidance from birds. Sounds that are modulated over time according to frequency and/or amplitude may elicit a higher degree of perceived urgency from the listener than constant, unchanging, or broadband tones (*Catchpole, McKeown & Withington, 2004*), such as the modulation of emergency sirens as an example familiar to humans. As a result, conspicuous signals with shifting sound properties, such as an oscillating frequency range, may be more salient as collision deterrents for birds.

Here, we evaluated the flight responses of zebra finches (*Taeniopygia guttata*) exposed to four sound treatments while flying through an outdoor flight corridor containing a visible

obstacle. We designed sound signals that varied in frequency ranges (4–6 and 6–8 kHz) and sound shapes (broadband noise within frequency ranges or frequency-modulated oscillations between these range limits) to determine whether particular sound properties more effectively elicited collision avoidance behaviors from the birds. We hypothesized that higher frequency (*i.e.*, 6–8 kHz) and frequency-modulated sound signals are the most effective signals because they are most easily detectable above the lower frequency background noise birds experience while in flight and evoke the most urgent avoidance responses. Because of this, we predicted that birds subjected to such signals would reduce their velocity, increase the distance between themselves and the flight obstacle, and adjust the trajectory of their flight sooner than birds exposed to other types of sound signals, all of which are behaviors that would contribute to a reduction in the risk of a harmful collision.

## MATERIALS & METHODS

### Study system

We used 25 captive, domesticated zebra finches housed in free-flight conditions in an indoor aviary (approximately 3.0 × 3.0 × 2.5 m; 18:6 light:dark photoperiod; temperature range 21–27 °C) in Williamsburg, Virginia. Birds were given ad libitum access to Volkman Avian Science seed mix, drinking water, bathing water, and perches. The zebra finches were bred on-site for use in this and other studies, from descendants of domesticated finches obtained through commercial dealers decades prior. Our euthanasia procedure involved instantaneous decapitation with sharp scissors when animals obtained unsurvivable injuries that caused pain or inhibited locomotion; however, this was not necessary for any of these 25 subjects, as all remained in good health throughout the duration of the experiments. At the conclusion of these experiments, all animals were returned to the breeding colony to produce more experimental subjects for future research.

Zebra finches are a suitable study system because their hearing sensitivities are similar to many other songbirds and they are more easily held in captivity than wild birds (*Dooling, 2002*; *Griffith & Buchanan, 2010*). We also know their flight responses in our testing tunnel are similar to those of a wild-caught species, brown-headed cowbirds (*Molothrus ater*) (*Swaddle et al., 2020*). We collected repeated measures of flights from each bird, identifiable by unique color combinations of plastic leg bands, in order to account for possible among-individual variation in flight behaviors. All procedures were approved by the William & Mary Institutional Animal Care and Use Committee (IACUC-2019-09-22-13861-jpswad).

### Sound treatments

To test the effectiveness of different acoustic deterrents to birds in flight, we created sound signals from all combinations of two frequency ranges (4–6 and 6–8 kHz) and two sound shapes (band and oscillations, defined below) (Fig. 1) using online software from WavTones (*Pigeon, 2019*) and AudioCheck (*Pigeon, 2018*). We selected the 4–6 kHz frequency to partially overlap with the documented peak auditory sensitivity of birds (*Dooling, 2002*), while the 6–8 kHz frequency lies beyond this range but has the potential to be more detectable as it is less likely to be masked by ambient sounds generated by the birds' motion or anthropogenic activities. We designed ''band'' treatment files to

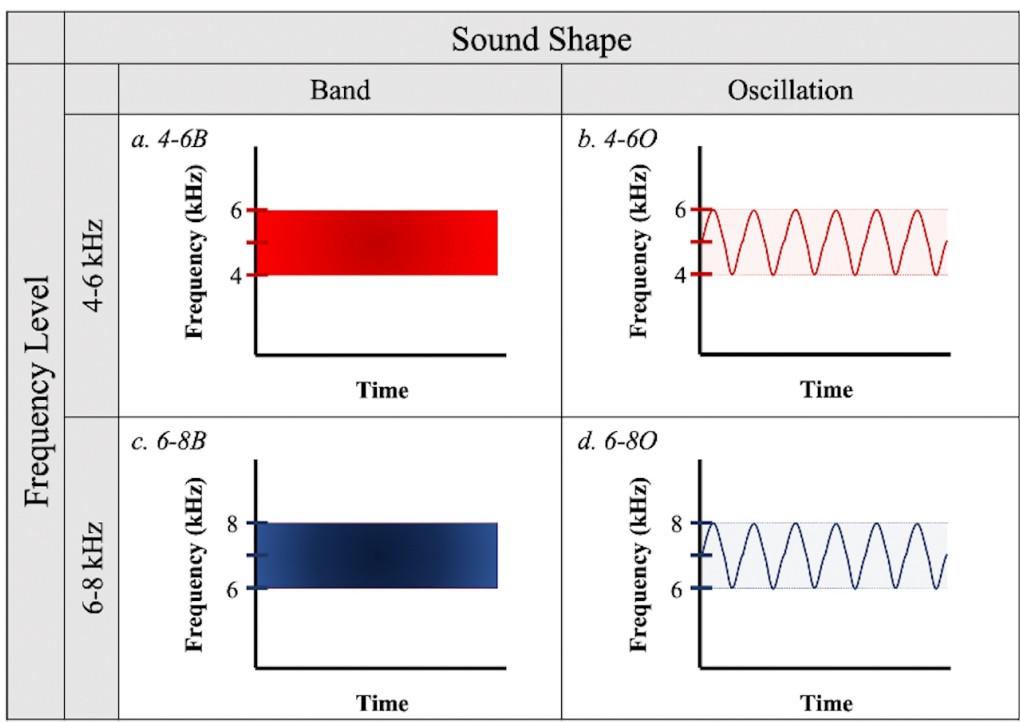

**Figure 1** **Sound treatments.** Four sound signals were created from all possible combinations of two frequency levels (4–6 kHz or 6–8 kHz) and two sound shapes ("Band" or "Oscillation").

contain a continuous spectrum of sound waves within their respective frequency ranges (*i.e.*, bandpass filtered white noise), while "oscillations" contained frequency modulations between the upper and lower limits of the frequency ranges, with only one pitch played at a time. The factorial combination of frequency ranges and sound shapes resulted in four acoustic treatments: (a) 4–6 kHz band (4-6B); (b) 4–6 kHz oscillation (4-6O); (c) 6–8 kHz band (6-8B); and (d) 6–8 kHz oscillation (6-8O) (Fig. 1).

## Flight trials

We performed flight trials from June 1 through July 30, 2020 between the hours of 0700 and 1200. One at a time, we released birds from a dark tunnel leading into an outdoor, naturally lit flight corridor (Fig. 2). At 3.6 m into the outdoor flight corridor (5.6 m total from the release point), we hung a black tarp (1 m wide) from ceiling-to-ground to present an obstacle in the birds' flight path. We placed a highly directional speaker (Holosonics AS-168i; 40 × 20 cm) directly adjacent to the front side of this obstacle, facing the dark release tunnel at a height of 1.2 m. In treatment flights, a sound signal played from this speaker at an amplitude of 85 dBA SPL measured by a sound meter (Galaxy Audio CM-130) at the emergence point of the dark tunnel (3.6 m from the speaker). The speaker emitted a highly directional sound field that was approximately constant in amplitude at all distances in the frontal direction of the speaker. The sound beam pointed directly toward the release point of the bird. The signal was initiated seconds prior to the release of the bird and played

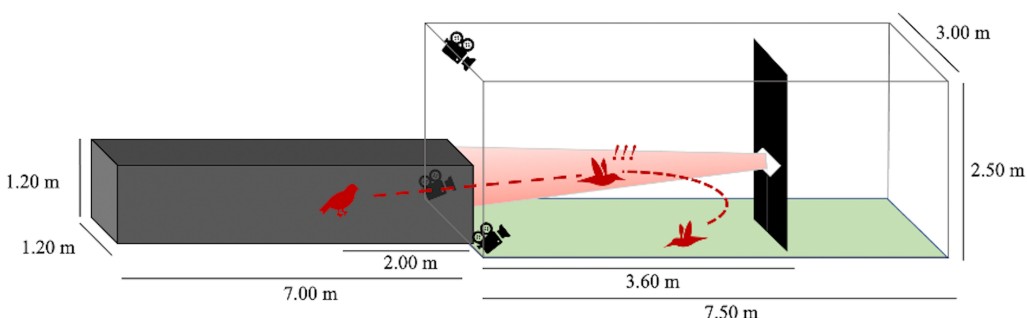

**Figure 2** **Flight corridor schematic.** Birds were released inside a dark tunnel (7.00 × 1.20 × 1.20 m) leading into an outdoor corridor (7.50 × 3.00 × 2.50 m). A tarp hanging ceiling-to-floor acted as a potential collision hazard. A speaker was positioned adjacent to the tarp so that the resultant sound beam (in treatment flights) was directed at the dark tunnel. The bird's flight pattern was recorded after its emergence from the dark tunnel using three Go-Pro cameras.

for the entire duration of the bird's flight. In control flights, no sound played from the speaker, but the speaker remained in place.

We measured flight behaviors from recordings on three Go-Pro Hero 7 Black video cameras (60 fps, 1440 resolution, 4:3 aspect ratio, linear shooting mode) arranged at staggered heights and angles to provide multiple recorded perspectives of the scene. This aided in recreating three-dimensional flight paths (see *Flight Digitization*). We regarded flights as complete when the bird changed direction by more than 90 degrees relative to the obstacle, flew past the obstacle, or landed on the ground within the outdoor flight corridor.

Each bird was exposed to all four acoustic treatments in a randomized order, and each treatment flight occurred within 24–48 h of a preceding control flight, for a total of eight flights for each bird. Birds had 5–7 days rest in their home aviary between consecutive control-treatment pairings. Pairing treatments with repeated control flights allowed us to monitor whether any within-individual changes in flight patterns resulted from acoustic treatments versus change in behavior due to the passage of time and/or repeated exposure to the flight corridor, as it is plausible that habituation to the experimental setup could occur over the course of the repeated measures. All analyses comparing treatments are made using the birds as controls for themselves at every time point, with treatment flight metrics adjusted by the immediately preceding control metrics (described in **Statistical Analyses**). Therefore, even if some degree of habituation did occur due to repeated exposure to the flight tunnel, our results captured the differences in behavior of birds exposed to sound signals relative to their own behavior at the same time point and approximately the same level of prior experience with the flight corridor.

We included a flight in analyses if the bird flew at least 2.5 m and appeared within the field of view of at least two of the three cameras, approximately 0.5 m beyond the end of the dark release tunnel. At this distance the bird had flown far enough to interact with the obstacle and sound signal (in treatment flights). Birds that failed up to two of their eight flights were given one month of rest before being exposed again to the missing control-treatment pairings and were retained in the study if these additional flights were
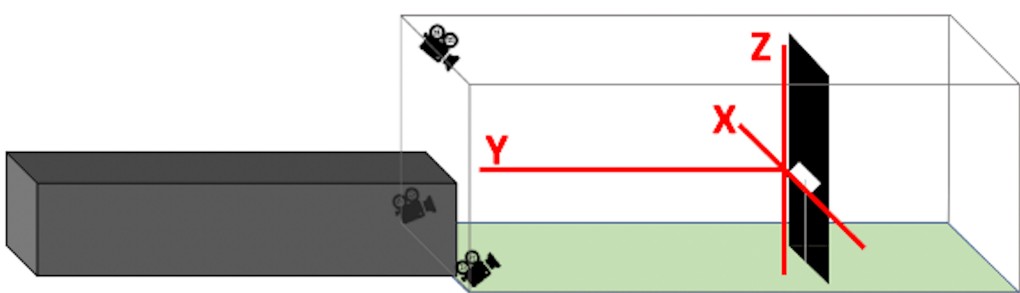

**Figure 3 Positions of *X*-, *Y*-, and *Z*-axes within flight corridor.** The *X*-axis spans from side to side, the *Y*-axis spans down the length of the corridor, and the *Z*-axis spans floor to ceiling. The three axes intersect at the center of the speaker to form the origin.

successful ($N = 8$). Birds that failed more than two flights were removed from the study and none of their data were used in analyses ($N = 6$). Nineteen birds successfully completed all eight (four control-treatment pairs) flights and were retained in analyses.

## Flight digitization

Using Argus software (*Jackson et al., 2016*), we synchronized and calibrated recordings from the three camera angles and manually digitized each bird's position in every frame of its flight duration from all three camera views. We synchronized the cameras with a combination of auditory and visual cues. After starting the recording on all three cameras and before each series of flight trials, we played a series of beeping tones through three walkie talkies that were each positioned within 0.05 m of each camera. We then turned a flashlight on and off, positioned such that the flash was visible on all cameras. For the calibration process, we maneuvered a 0.46 m wand with brightly colored spheres affixed to either end around the entire volume of the flight corridor (*Theriault et al., 2014*). In Argus, we plotted points on the centroid of each sphere for 40 to 60 frames. Providing these paired points from multiple perspectives enabled Argus to compute how the overlap of various pixel locations corresponded to actual spatial distances within the tunnel. In order to establish the *X*-, *Y*-, and *Z*-axes of the active flight space, where the origin was defined as the center of the speaker (Fig. 3), we recorded a right-angled three-dimensional PVC structure held in the view of all three cameras, level with the speaker, and digitized its extremities.

After synchronization and calibration of all videos, we digitized each individual bird's flights from the initial frame it appeared on at least two of the three cameras until the relevant portion of the flight had been completed (*i.e.*, the bird changed direction by more than 90 degrees relative to the obstacle, flew past the obstacle, or landed on the ground). In Argus, we combined digitized points from each video for each flight to produce three-dimensional locations of each bird on each frame of each video. We smoothed these *X*-, *Y*-, and *Z*-coordinates with a moving window average of eleven frames (the instantaneous coordinate as well as coordinates from the five preceding and five following frames) in order to reduce the influence of digitization error on our calculations of flight metrics.

**Table 1  Intermediate calculations.** These formulae were used to determine the birds velocity, distance, and changes in flight trajectory at each frame using their $x$-, $y$-, and $z$- coordinates. These instantaneous values were then used to calculate the flight metrics described in Table 2.

| Description | Formulae and notes |
|---|---|
| Instantaneous velocity (m/s) at frame $n$ | $v_n = \sqrt{(x_n - x_{n-1})^2 + (y_n - y_{n-1})^2 + (z_n - z_{n-1})^2} \quad * \quad 60$ <br> The vector distance (m) between two consecutive positions is multiplied by the frame rate (60 fps) to achieve instantaneous velocity in m/s. |
| Distance (m) at frame $n$ | **(a)** With respect to the obstacle at $X = 0$ and $Y = 0$ (excluding the $Z$ plane, as the obstacle occupies all possible $Z$ coordinates): <br> $d_{n_{obstacle}} = \sqrt{x_n^2 + y_n^2}$ <br> **(b)** With respect to the speaker at $X=0$, $Y=0$, and $Z=0$: <br> $d_{n_{speaker}} = \sqrt{x_n^2 + y_n^2 + z_n^2}$ |
| Change in flight trajectory at frame $n$ | **(a) Curvature (rad/m):** <br> $\kappa = \dfrac{\left\lvert x'z'' - z'x'' \right\rvert}{x'^2 + z'^2}$ <br> **(b) Angle of inflection (rad):** <br> $\Theta_n = \cos^{-1} \dfrac{\left(x_{n+1_{expected}} - x_n\right) * \left(x_{n+1_{actual}} - x_n\right) + \left(z_{n+1_{expected}} - z_n\right) * \left(z_{n+1_{actual}} - z_n\right)}{\sqrt{\left(x_{n+1_{expected}} - x_n\right)^2 + \left(z_{n+1_{expected}} - z_n\right)^2} * \sqrt{\left(x_{n+1_{actual}} - x_n\right)^2 + \left(z_{n+1_{actual}} - z_n\right)^2}}$, <br> where $x_{n+1_{expected}} = x_n + (x_n - x_{n-1})$ and $z_{n+1_{expected}} = z_n + (z_n - z_{n-1})$ <br> Both calculations exclude movement in the $Y$ direction, which dilutes any meaningful changes in trajectory by including the bird's forward motion. The greatest change of flight trajectory was determined as the frame during which both curvature and angle of inflection were maximized. |

## Metric calculation

Prior to extracting quantitative metrics from the bird flights, we scored each flight qualitatively in terms of the general flight patterns, which helped us define appropriate metrics that described differences among the flights (Table S1). Because most birds flew a moderate or long distance into the day lit flight tunnel and zig-zagging flight patterns were uncommon, finding each bird's maximum instantaneous angle of inflection appropriately captured the moment in the flight during which the bird made the greatest adjustment to its overall trajectory relative to continuing down a straight path towards the obstacle.

For each frame of every flight, we calculated the instantaneous velocity (m/s), distance from both the obstacle and the speaker (m), and change in flight trajectory with respect to the position that would be predicted if the bird continued flying in a straight line from its previous coordinate (Table 1). We then used these intermediate calculations to compute seven metrics of flight behaviors, defined in Table 2.

We summarized velocity as (1) the within-flight change in velocity from the first third to the final third of the flight and (2) the average velocity over the entire course of the flight. These metrics, respectively, captured any acceleration or deceleration that occurred as the bird's flight progressed, as well as whether the overall flight velocity differed between treatments and controls or among different sound treatments. We determined the minimum distance between the bird and both (3) the obstacle and (4) the speaker, as well as the distance between the bird and both (5) the obstacle and (6) the speaker when the greatest change in flight trajectory occurred, which generated metrics of how close birds

**Table 2  Flight metrics.** Seven metrics of collision avoidance were computed from birds three-dimensional coordinates, related to flight velocity, distance from speaker and collision hazard, and change in flight trajectory.

| Metric description | Calculation method |
| --- | --- |
| 1. Within-flight change in velocity (m/s) | Average velocity in final third of flight minus average velocity in first third of flight |
| 2. Average velocity (m/s) | Average velocity over the entire course of the flight |
| 3. Minimum distance from obstacle (m) | Smallest distance measurement between the bird and the obstacle |
| 4. Minimum distance from speaker (m) | Smallest distance measurement between the bird and the speaker |
| 5. Distance from obstacle when bird makes greatest adjustment in flight trajectory (m) | Distance between the bird and the obstacle at frame when change in flight trajectory is greatest (both curvature and angle of inflection are maximized) |
| 6. Distance from speaker when bird makes greatest adjustment in flight trajectory (m) | Distance between the bird and the speaker at frame when change in flight trajectory is greatest |
| 7. Proportion of flight completed when bird makes greatest adjustment in flight trajectory | Frame number at which greatest change in flight trajectory occurs divided by total number of frames in flight |

came to experiencing collisions under different conditions and where in the course of the flight they altered their flight to avoid the hazard. Finally, we determined the proportion of the total flight that had occurred when this maximum change in flight trajectory occurred (7). This generated a relative metric of how early/late in a flight the birds adjusted their trajectory.

We inferred relative collision risk from these metrics. We interpreted that flights in which birds moved at a slower velocity, maintained a greater distance from obstacles, and/or adjusted their trajectory farther away from hazards were most likely to result in avoidance of a collision. Furthermore, assessing distance from the speaker and from the obstacle separately allowed insight into whether any avoidance behaviors resulted in navigation away from merely the source of the signal (*i.e.*, flying above or below speaker level but still in line with the obstacle) or also involved successful navigation around the physical flight hazard.

## Statistical analyses

We performed all statistical analyses using R version 3.6.3. We confirmed that all data met assumptions of normality using Shapiro–Wilk tests. To determine whether flight behaviors differed according to whether or not a treatment was used, the frequency level of the sound signal, and/or the sound shape, we performed three-factor Type III repeated measures analyses of variance (ANOVA) using the function "anova_test()" from the rstatix package (*Kassambara, 2021*). These calculations used mixed-effects models with bird ID as a random effect. Within-individual factors included treatment (control *vs.* treatment flights for each bird), frequency (4–6 *vs.* 6–8 kHz), and sound shape (band *vs.* oscillation), as well as the interaction between frequency and sound shape. Initially, we performed this analysis separately for each of the seven flight metrics.

We also computed a principal components analysis (PCA) to determine whether flight metrics were correlated and subsequently associated with any of the sound signals. To do

this, we transformed the data by subtracting the metric calculations of each control flight from their paired treatment flights, thus providing a treatment measurement relative to baseline flight behavior. We performed a PCA that included all seven adjusted flight metrics (treatment minus control) as input variables using the "princomp()" function from base R. We analyzed differences in PC1 and PC2 scores separately in response to the four sound signals through two-way repeated measures ANOVAs with frequency and sound shape as within-individual factors. Measurements are reported as means ± SEM.

## RESULTS

### Sound treatments *vs.* control flights

Birds maintained a greater minimum distance from the obstacle during treatment flights compared with control flights ($F_{1,18} = 21.40$, $p = 0.0002$). Specifically, birds kept about 50% more distance from the obstacle (black tarp) during sound treatment flights ($2.12 \pm 0.13$ m) than in controls ($1.41 \pm 0.12$ m). Sound treatments also caused birds to maintain a greater minimum distance from the speaker ($F_{1,18} = 28.59$, $p = 0.00004$). Birds flew 39% further from the speaker in treatment flights ($2.53 \pm 0.13$ m) compared with in control flights ($1.82 \pm 0.11$ m).

Birds also made the greatest adjustment to their flight trajectory at a further distance away from both the obstacle ($F_{1,18} = 17.70$, $p = 0.0005$) and the speaker ($F_{1,18} = 21.65$, $p = 0.0002$) during treatment flights compared with controls. This adjustment occurred at an average of $2.89 \pm 0.11$ m from the obstacle and $3.30 \pm 0.11$ m from the speaker in all treatment flights compared with means of $2.33 \pm 0.13$ m and $2.68 \pm 0.13$ m, respectively, in controls. Thus, the sound signals elicited the greatest change in flight trajectory about 26% further away from the obstacle and about 22% further away from the speaker.

There was no difference between control and treatment flights for the within-flight change in velocity ($F_{1,18} = 0.41$, $p = 0.53$), average velocity ($F_{1,18} = 3.02$, $p = 0.10$), or the proportion of flight completed at the instant when the bird made the greatest adjustment in flight trajectory ($F_{1,18} = 2.55$, $p = 0.13$) (Fig. 4, Tables 3 and 4).

### Comparisons among sound signal types

For each of the seven metrics of flight, we found no statistically supported differences in flight behavior according to frequency level (*i.e.*, 4–6 *vs.* 6–8 kHz), sound shape (*i.e.*, band *vs.* oscillation), or the interaction of the two (Table 4). However, after inspecting means and 95% confidence intervals of treatment flights relative to controls (Table 3 and Fig. 5), we observed that flight responses to the 4-6O signal and to the 6-8B signal resulted in behaviors associated with a greater probability of collision avoidance. Birds exposed to the 4-6O signal maintained the greatest distance from both the obstacle (5C) and the speaker (5D), adjusted their flight trajectories further away from both the obstacle (5E) and the speaker (5F), and made this maximum change in their flight trajectory earlier than birds exposed to all other sound signals (5G). Birds exposed to the 6-8B signal decreased their velocity the most over the course of the flight (5A) and maintained a greater distance from both the obstacle and the speaker (5C–5D).

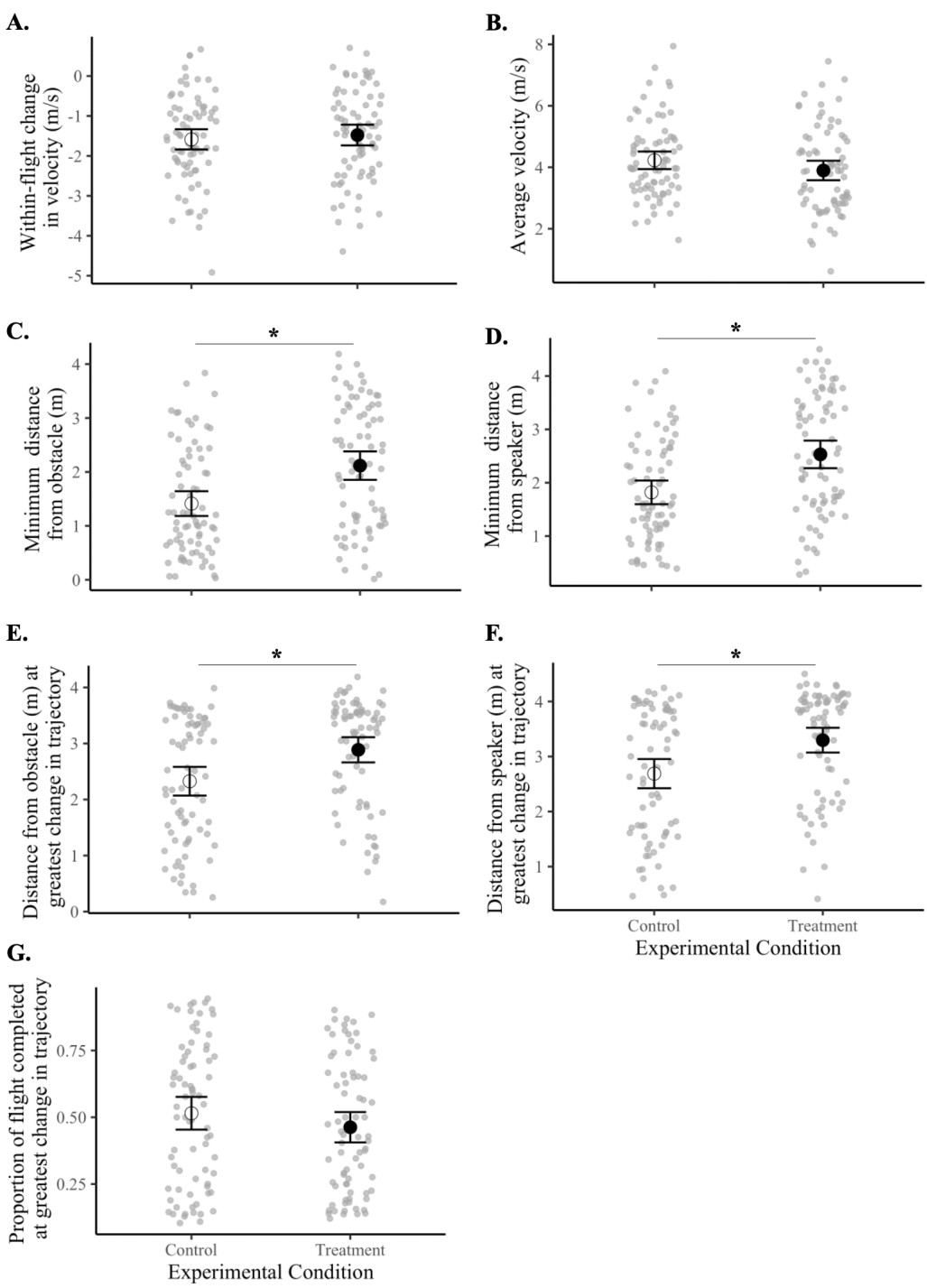

**Figure 4  Flight behaviors of birds under control *vs.* treatment conditions.** Data shown are means (±95% CI) of all control flights (open circle) and of all treatment flights (filled circle). Magnitudes and units of *y*-axes differ from panel to panel. Asterisks (*) indicate a significant difference between control and treatment flights ($p < 0.05$). Panels represent the (A) within-flight change in velocity (m/s), (B) average velocity (m/s), (C) minimum distance from the obstacle (m), (D) minimum distance from the speaker (m), (E) distance from obstacle (m) at greatest change in flight trajectory, (F) distance from speaker (m) at greatest change in flight trajectory, and (G) proportion of flight completed at greatest change in flight trajectory.

**Table 3  Flight behavior summary statistics according to experimental conditions and sound signal attributes.** Data show means ± SEM and 95% CI (given in brackets as "[lower boundary, upper boundary]"). Columns A and B group together all control flights and all treatment flights, respectively. Columns C–F provide the adjusted (treatment minus control) data for each type of sound signal separately.

| Flight metric | (A) All control flights | (B) All treatment flights | (C) 4-6 B treatment flights (adjusted by controls) | (D) 4-6 O treatment flights (adjusted by controls) | (E) 6-8 B treatment flights (adjusted by controls) | (F) 6-8 O treatment flights (adjusted by controls) |
|---|---|---|---|---|---|---|
| 1. Within-flight change in velocity (m/s) | −1.59 ± 0.13 m/s [−1.84, −1.33] | −1.48 ± 0.13 m/s [−1.74, −1.22] | 0.24 ± 0.33 m [−0.46, 0.94] | 0.34 ± 0.39 m [−0.47, 1.15] | −0.41 ± 0.31 m [−1.06, 0.25] | 0.26 ± 0.35 m [−0.47, 0.98] |
| 2. Average velocity (m/s) | 4.23 ± 0.14 m/s [3.94, 4.51] | 3.90 ± 0.16 m/s [3.58, 4.21] | 0.02 ± 0.30 m [−0.618, 0.657] | −0.53 ± 0.44 m [−1.45, 0.40] | −0.70 ± 0.30 m [−1.32, −0.07] | −0.13 ± 0.35 m [−0.85, 0.60] |
| 3. Minimum distance from obstacle (m) | 1.41 ± 0.12 m [1.18, 1.64] | 2.12 ± 0.13 m [1.85, 2.38] | 0.34 ± 0.32 m [−0.32, 1.01] | 0.86 ± 0.33 m [0.17, 1.55] | 1.14 ± 0.27 m [0.58, 1.71] | 0.47 ± 0.29 m [−0.13, 1.08] |
| 4. Minimum distance from speaker (m) | 1.82 ± 0.11 m [1.60, 2.04] | 2.53 ± 0.13 m [2.27, 2.79] | 0.40 ± 0.26 m [−0.14, 0.95] | 0.94 ± 0.33 m [0.25, 1.63] | 0.98 ± 0.24 m [0.48, 1.49] | 0.52 ± 0.27 m [−0.05, 1.09] |
| 5. Distance from obstacle when bird makes greatest adjustment in flight trajectory (m) | 2.33 ± 0.13 m [2.07, 2.58] | 2.89 ± 0.11 m [2.66, 3.11] | 0.03 ± 0.31 m [−0.61, 0.67] | 1.12 ± 0.29 m [0.50, 1.74] | 0.67 ± 0.37 m [−0.10, 1.44] | 0.43 ± 0.27 m [−0.14, 0.99] |
| 6. Distance from speaker when bird makes greatest adjustment in flight trajectory (m) | 2.68 ± 0.13 m [2.42, 2.95] | 3.30 ± 0.11 m [3.07, 3.52] | 0.06 ± 0.29 m [−0.55, 0.67] | 1.28 ± 0.27 m [0.71, 1.85] | 0.64 ± 0.36 m [−0.12, 1.39] | 0.45 ± 0.27 m [−0.12, 1.02] |
| 7. Proportion of flight completed when bird makes greatest adjustment in flight trajectory | 0.52 ± 0.03 [0.45, 0.58] | 0.46 ± 0.03 [0.41, 0.52] | 0.07 ± 0.07 m [−0.07, 0.21] | −0.17 ± 0.09 m [−0.35, 0.02] | −0.04 ± 0.09 m [−0.22, 0.15] | −0.07 ± 0.07 m [−0.22, 0.07] |

## Principal components analysis

PCA loadings are shown in Table 5, wherein components 1 and 2 account for 81% of the observed variance. Based on the directionality of these loadings, component 1 (PC1) is positively associated with flights in which the birds flew more quickly, came closer to the obstacle and the speaker, and adjusted their angle of flight when in closer proximity to the obstacle and the speaker in the treatment than in the control. Therefore, a decreasing score for PC1 indicates increased collision avoidance as birds flew more slowly and further away from the collision threat. PC2 is positively associated with flights in which birds accelerated, flew more quickly, and adjusted their angle earlier and further from the obstacle and speaker in the treatment than in the control. Therefore, an increasing score for PC2 would indicate responses where birds adjusted the direction of their flight early and accelerated thereafter. We interpreted this to be an additional way in which birds could avoid a collision.

PC1 and PC2 scores did not differ in response to frequency levels (PC1: $F_{1,18} = 0.19$, $p = 0.67$; PC2: $F_{1,18} = 0.83$, $p = 0.38$), sound shapes (PC1: $F_{1,18} = 0.74$, $p = 0.40$; PC2:
**Table 4 Statistical comparisons of flight behaviors according to experimental conditions and sound signal attributes.** The seven flight metrics (described in Table 2) were compared using three-factor Type III ANOVA to determine whether flight behaviors differed according to the frequency level of the sound signal and/or the sound shape. F statistics, $p$-values, and generalized effect sizes ($\eta_G^2$) are given for each factor. Values are reported to three significant digits with the exception of those smaller than 0.01, which are denoted by "<0.01" or by "<<0.01" if the difference is greater than one order of magnitude. Significant comparisons are shown with bolded $p$-values and a single asterisk (*).

| Flight metric | Comparisons between control and treatment flights | Comparisons among sound signals by frequency level | Comparisons among sound signals by sound shape | Comparisons among sound signals by interaction of frequency and shape |
|---|---|---|---|---|
| 1. Within-flight change in velocity (m/s) | $F_{1,18} = 0.41$<br>$p = 0.53$<br>$\eta_G^2 < 0.01$ | $F_{1,18} = 0.80$<br>$p = 0.38$<br>$\eta_G^2 = 0.01$ | $F_{1,18} = 3.15$<br>$p = 0.09$<br>$\eta_G^2 = 0.01$ | $F_{1,18} = 0.54$<br>$p = 0.47$<br>$\eta_G^2 < 0.01$ |
| 2. Average velocity (m/s) | $F_{1,18} = 3.02$<br>$p = 0.10$<br>$\eta_G^2 = 0.02$ | $F_{1,18} = 0.45$<br>$p = 0.51$<br>$\eta_G^2 << 0.01$ | $F_{1,18} < 0.01$<br>$p = 0.96$<br>$\eta_G^2 << 0.01$ | $F_{1,18} = 1.37$<br>$p = 0.26$<br>$\eta_G^2 = 0.01$ |
| 3. Minimum distance from obstacle (m) | $F_{1,18} = 21.4$<br>$\boldsymbol{p} <<\mathbf{0.01}$ (*)<br>$\eta_G^2 = 0.10$ | $F_{1,18} = 0.60$<br>$p = 0.45$<br>$\eta_G^2 < 0.01$ | $F_{1,18} = 0.11$<br>$p = 0.75$<br>$\eta_G^2 < 0.01$ | $F_{1,18} = 2.38$<br>$p = 0.14$<br>$\eta_G^2 = 0.02$ |
| 4. Minimum distance from speaker (m) | $F_{1,18} = 28.6$<br>$\boldsymbol{p} <<\mathbf{0.01}$ (*)<br>$\eta_G^2 = 0.11$ | $F_{1,18} = 0.10$<br>$p = 0.76$<br>$\eta_G^2 << 0.01$ | $F_{1,18} = 0.04$<br>$p = 0.85$<br>$\eta_G^2 << 0.01$ | $F_{1,18} = 1.94$<br>$p = 0.18$<br>$\eta_G^2 = 0.01$ |
| 5. Distance from obstacle when bird makes greatest adjustment in flight trajectory (m) | $F_{1,18} = 17.7$<br>$\boldsymbol{p} <<\mathbf{0.01}$ (*)<br>$\eta_G^2 = 0.07$ | $F_{1,18} = 0.01$<br>$p = 0.91$<br>$\eta_G^2 << 0.01$ | $F_{1,18} = 2.36$<br>$p = 0.14$<br>$\eta_G^2 = 0.01$ | $F_{1,18} = 2.54$<br>$p = 0.13$<br>$\eta_G^2 = 0.03$ |
| 6. Distance from speaker when bird makes greatest adjustment in flight trajectory (m) | $F_{1,18} = 21.7$<br>$\boldsymbol{p} <<\mathbf{0.01}$ (*)<br>$\eta_G^2 = 0.08$ | $F_{1,18} = 0.27$<br>$p = 0.61$<br>$\eta_G^2 << 0.01$ | $F_{1,18} = 3.36$<br>$p = 0.08$<br>$\eta_G^2 = 0.02$ | $F_{1,18} = 3.16$<br>$p = 0.09$<br>$\eta_G^2 = 0.03$ |
| 7. Proportion of flight completed when bird makes greatest adjustment in flight trajectory | $F_{1,18} = 2.55$<br>$p = 0.13$<br>$\eta_G^2 = 0.01$ | $F_{1,18} < 0.01$<br>$p = 0.94$<br>$\eta_G^2 << 0.01$ | $F_{1,18} = 2.78$<br>$p = 0.11$<br>$\eta_G^2 = 0.02$ | $F_{1,18} = 1.31$<br>$p = 0.27$<br>$\eta_G^2 = 0.01$ |

$F_{1,18} = 3.32$, $p = 0.09$), or the interaction of frequency and sound shape (PC1: $F_{1,18} = 3.20$, $p = 0.09$; PC2: $F_{1,18} = 0.0004$, $p = 0.99$). We observed a non-statistically-supported trend indicating that birds exposed to the 4-6O signal had the lowest average scores for PC1 and the highest average scores for PC2 (Fig. 6), thus maximizing both types of collision-avoidant behaviors. Birds also tended to have lower average scores for both PC1 and PC2 in response to the 6-8B signal than to the remaining two signals, though this pattern was not statistically supported.

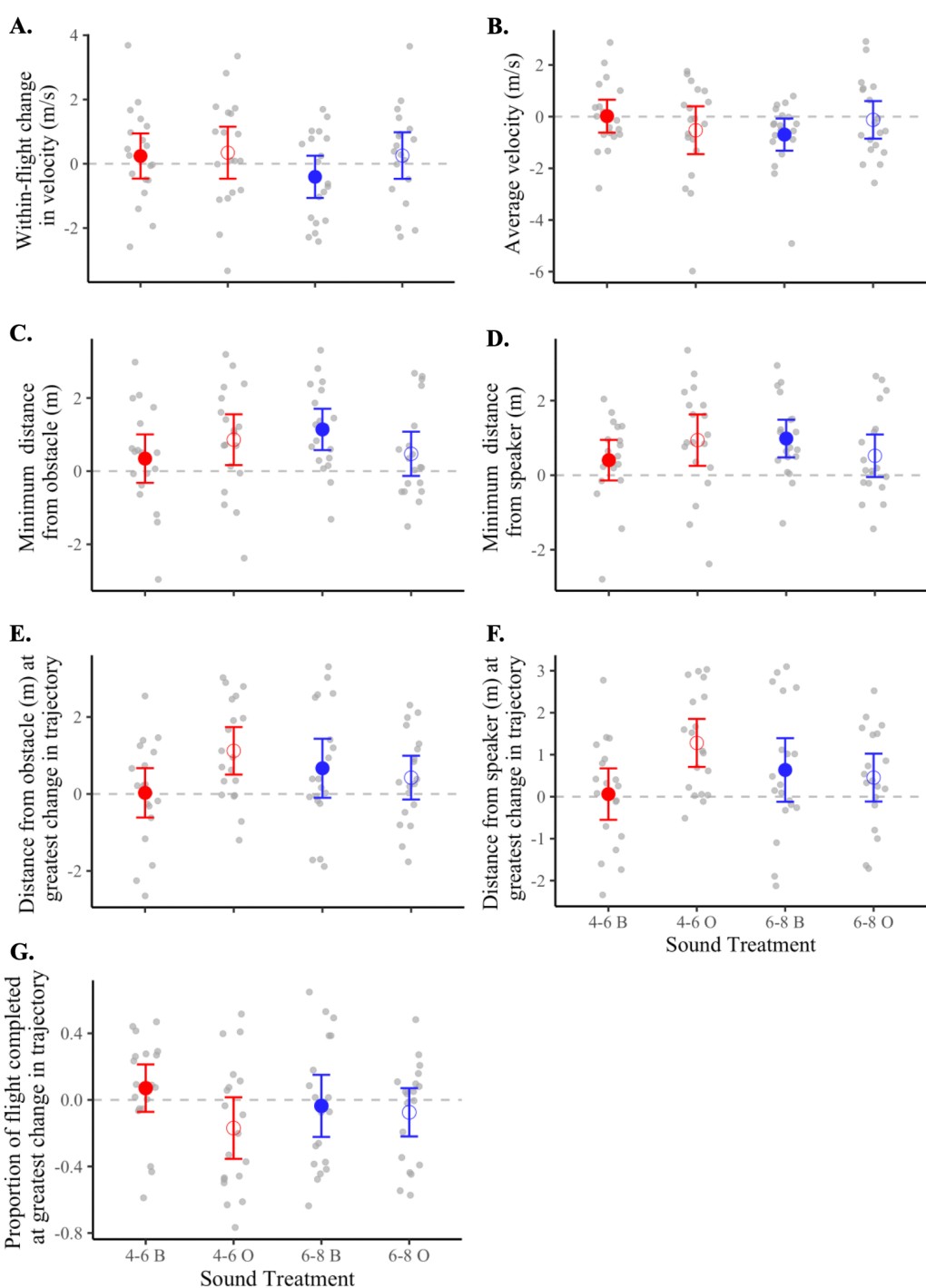

**Figure 5 Flight metrics of birds in response to different sound treatments.** Data shown are means (±95% CI) of treatment response minus control response for each bird. Dotted lines indicate the threshold at which there is no difference in the metric of interest between treatment and control flights. Magnitudes and units of $y$-axes differ from panel to panel. Treatments: 4–6 kHz in red, 6–8 kHz in blue; band signals given in closed circles and abbreviated "B", oscillating signals given in open circles and abbreviated "O". (continued on next page...)

**Figure 5 (…continued)**
Panels represent the (A) within-flight change in velocity (m/s), (B) average velocity (m/s), (C) minimum distance from the obstacle (m), (D) minimum distance from the speaker (m), (E) distance from obstacle (m) at greatest change in flight trajectory, (F) distance from speaker (m) at greatest change in flight trajectory, and (G) proportion of flight completed at greatest change in flight trajectory.

**Table 5 Flight metric loadings in principal components 1 and 2.** Positive loadings are shown in green and negative loadings are shown in red.

| Metric description | PC 1 loading | PC 2 loading |
|---|---|---|
| 1. Within-flight change in velocity (m/s) | 0.167 | 0.684 |
| 2. Average velocity (m/s) | 0.398 | 0.515 |
| 3. Minimum distance from obstacle (m) | −0.473 | – |
| 4. Minimum distance from speaker (m) | −0.423 | – |
| 5. Distance from obstacle when bird makes greatest adjustment in flight trajectory (m) | −0.463 | 0.342 |
| 6. Distance from speaker when bird makes greatest adjustment in flight trajectory (m) | −0.440 | 0.370 |
| 7. Proportion of flight completed when bird makes greatest adjustment in flight trajectory | – | −0.106 |

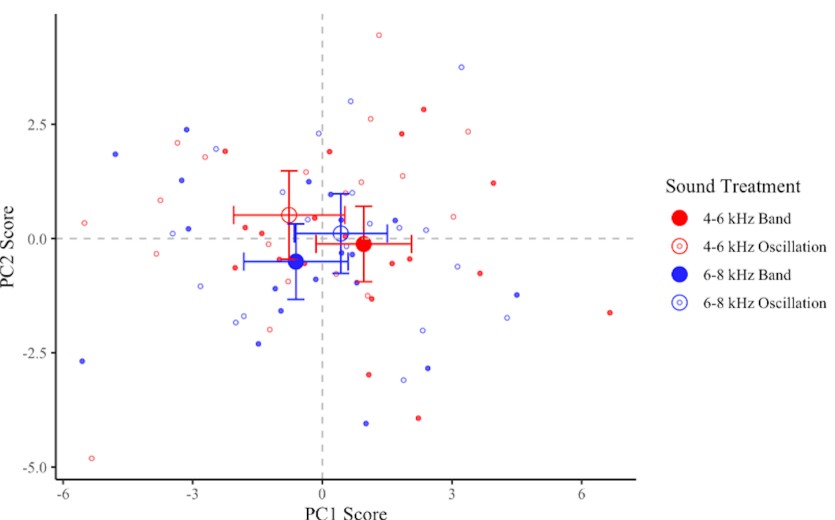

**Figure 6 Plot of PC1 and PC2 for each flight.** Data shown are means ±95% CI of treatment minus control for each bird in each treatment. Treatments: 4–6 kHz in red, 6–8 kHz in blue; band signals given in closed circles, oscillating signals given in open circles. More negative PC1 scores and more positive PC2 scores are interpreted as collision-avoidant flight behaviors.

# DISCUSSION

We found that the acoustic signals caused birds to maintain a greater distance from potential collision hazards and to adjust their trajectories before coming close to these hazards, as compared to flights during control conditions. Behavioral responses did not differ significantly between different acoustic signals, implying that a variety of different

sounds may elicit similar collision-avoidance behaviors from birds. From the range of signals we tested, there were trends indicating that a signal whose sound was modulated in frequency between 4 to 6 kHz (*i.e.*, 4-6O signal) tended to elicit slightly stronger changes in flight behaviors, but due to the lack of statistical significance, we are cautious not to conclusively interpret biological meaning from these differences.

The fact that birds maintained a greater distance and adjusted their flight trajectory further away from both the speaker and the obstacle during treatment flights suggests that the birds adjusted their flight behavior in the tunnel earlier when signals were used, allowing them time to navigate away from the object in their path. This finding may also explain why the average flight velocity did not differ between birds in treatments versus controls, as it may not have been necessary to slow down to avoid a collision when birds were already making other adjustments to their flight strategies to navigate around the hazard.

Although it is difficult to disentangle birds' avoidance of the obstacle with their avoidance of the speaker and sound beam, we do suspect that the sound avoidance was likely coupled with some degree of awareness of the physical obstacle itself. Some precedence for this conclusion is reported in *Swaddle & Ingrassia (2017)*, in which birds adjusted their flight behaviors when presented with objects-and-sound combined but did not do so when presented with a sound-only treatment. In our experimental setup, birds would have acoustically detected a decrease in sound level through any vertical or lateral deviations outside of the directional sound beam. If birds were solely avoiding the loud sound, it is plausible that some of them may still have continued to fly on track with a strike to the physical obstacle, which was both taller and wider than the directional sound field itself.

Measuring distance from the speaker valuably provided insight into their navigation away from the sound beam, while measuring distance from the obstacle allowed us to determine the extent to which this adjustment in flight behavior corresponded to a reduction in the risk of an actual collision. Had the results for these two flight behaviors not been quite so similar—*i.e.*, if sound was found to increase distance from the speaker but not from the obstacle (which would be possible if birds flew up, down, or slightly to the side while continuing towards the obstacle, as opposed to the larger and wider deviations in their flight paths we witnessed)—we instead would conclude that birds avoiding sound provided little reduction in their collision risk. As this was not the case, although we cannot conclusively suggest that birds were completely aware of the obstacle, we can demonstrate that the nature of their flight behaviors—whether from avoidance of a loud noise or awareness of a physical hazard—resulted in a reduced risk of collision, a desired result from a conservation perspective.

The effectiveness of these signals in a controlled setting suggests the potential for success in practical use. The finding that sound treatments in general elicit collision avoidance behaviors provides encouraging evidence that implementing many types of acoustic signals may increase birds' attention to their surrounding environments, reducing the risk of a fatal collision. Given the slight pattern that the 4-6O treatment elicited changes in flight behaviors most consistent with collision avoidance, we urge further study of frequency-modulated tones in field testing. To date, our research group has indicated that 4–6 kHz band sounds
can more effectively deviate migrating birds around communication towers than 6–8 kHz bands (*Boycott et al., 2021*). We predict that adding frequency modulation will make these sounds even more effective.

In addition to indicating which acoustic signals might be effective in reducing collision risk, our study underscores the value in using behavioral data to evaluate collision mitigation strategies. Most common metrics of collision risk are derived by collecting carcasses from hazardous landscapes, which may substantially undercount mortality from collisions due to the effects of scavenger removal (*Loss et al., 2019*; *Bracey et al., 2016*; *Kummer et al., 2016*). In addition, these methodologies fail to account for non-fatal collisions in which birds may endure physical damage in the aftermath of strikes that affects their survival despite not causing immediate mortality (*Boycott et al., 2021*). Assessing the consequences of bird strikes with behavioral data can better capture the costs to birds that survive collisions. Birds may also experience energetic tradeoffs even when they successfully avoid a collision (*Boycott et al., 2021*). Birds employ locomotive responses similar to antipredator behaviors in order to evade collisions (*Bernhardt et al., 2010*), and obstacle evasion typically involves adjusting the body angle and flapping more frequently to increase maneuverability (*Lin, Ros & Biewener, 2014*), which may prove costly, especially to migratory species. Considering adjustments in flight behavior could allow us to better interpret some of the threats posed to birds by manmade obstacles and of the innovations intended to reduce this risk. Furthermore, behavioral metrics may permit evaluation of intervention strategies to occur on a much shorter timescale, as it may be possible to record large amounts of behavioral data within days or weeks in comparison to the months or years required for fatality studies to reach sufficient sample sizes (*Boycott et al., 2021*).

This study reinforces a growing body of evidence that acoustic signals can increase the detectability of visible obstacles to birds (*Swaddle & Ingrassia, 2017*; *Boycott et al., 2021*), potentially reducing bird mortality and injury from collisions. Existing visual deterrent strategies, such as painting turbine blades black to increase visibility (*May et al., 2020*) or placement of lights on tower infrastructure (*Goller et al., 2018*), can be augmented by acoustic signals to further reinforce the probability of visual detection and collision evasion by birds. Utilizing acoustic signals should reduce the damage to humanmade structures caused by bird collisions in addition to reducing the collisions themselves, which is societally and economically desirable. Such technology may also permit the expansion of renewable wind energy with lessened disturbance to avifauna.

Importantly, although acoustic signals can be an effective collision deterrent in some settings, they likely are not the most appropriate technology to deploy in certain contexts. In human-adjacent areas, frequent use of conspicuous signals may create a nuisance effect to residents that undermines the practicality of their use. Similarly, sound signals may have unintended consequences on non-avian wildlife, possibly interfering with communication or inducing other forms of physiological stress (*Kight & Swaddle, 2011*); it is therefore essential to evaluate these potential externalities thoroughly before implementing acoustic signals. Additionally, there are some contexts in which sound is unlikely to reduce (or may even increase) the risk of a collision. Bird-window collisions, for example, are hypothesized to result from birds failing to perceive windows as solid obstacles, instead likely seeing

reflected vegetation or open space (*Klem, 2008*). Acoustic signals may therefore not achieve the intended collision prevention in contexts where the underlying cause of a collision is a perceptual deficit rather than failure to look straight ahead while flying. Furthermore, some visual cues—such as artificial light at night—actively attract birds (*Lao et al., 2020*), so implementing sound signals that draw their attention to these structures may increase the risk of a collision rather than act as a deterrent (*Swaddle & Ingrassia, 2017*). Under these circumstances, we do not recommend the use of acoustic signals and suggest that other anti-collision methodologies be further investigated as alternatives.

## CONCLUSIONS

We found that acoustic signals caused birds to maintain a greater distance from a physical obstacle and to adjust their flight trajectories early on in avoidance of a collision. Although we hypothesized that higher frequency oscillating sounds would be more detectable to birds in flight, we saw no statistical differences in birds' responses to different types of sounds, suggesting that a variety of different conspicuous signals may be able to elicit the behavioral changes necessary to avoid a collision. Our research additionally reinforces the value in measuring the flight behaviors of birds to assess collision risk and to evaluate the success of potential collision mitigation strategies. Continuing to develop, refine, and implement collision reduction technology such as acoustic signals will minimize the need to impose unrealistic constraints on our own development while also reducing the consequences of this development on wildlife.

## ACKNOWLEDGEMENTS

We thank Dr. Greg Conradi-Smith for invaluable input on the calculation of flight metrics. We also thank Dr. Brandon Jackson, whose feedback on experimental design and technological assistance with Argus was a tremendous asset. We are grateful to Dr. Dan Cristol and Dr. Matthias Leu for their comments throughout the conception and analysis of this project and for feedback on an early draft of this manuscript. We are indebted to the William & Mary Vertebrate Animal Care Unit—Shirley Mitchell, Gwen Carter, Kim Nicholson, and Alyster Carter—as well as Tim Boycott, Sally Mullis, and Tom Meier for contributing to the construction of the flight tunnel. We also thank Dr. Dany Garant, Dr. Miguel Ferrer, and two anonymous reviewers for their thoughtful consideration of our manuscript.

### Funding

This work was supported by the Virginia Space Grant Consortium, Virginia Society of Ornithology, Williamsburg Bird Club, Coastal Virginia Wildlife Observatory, and William & Mary Arts & Sciences. The funders had no role in study design, data collection and analysis, decision to publish, or preparation of the manuscript.

## Grant Disclosures

The following grant information was disclosed by the authors:

Virginia Space Grant Consortium.

Virginia Society of Ornithology.

Williamsburg Bird Club.

Coastal Virginia Wildlife Observatory, and William & Mary Arts & Sciences.

## Competing Interests

The authors declare that they have no competing interests.

## Author Contributions

- Robin G. Thady conceived and designed the experiments, performed the experiments, analyzed the data, prepared figures and/or tables, authored or reviewed drafts of the paper, and approved the final draft.
- Lauren C. Emerson performed the experiments, authored or reviewed drafts of the paper, and approved the final draft.
- John P. Swaddle conceived and designed the experiments, analyzed the data, authored or reviewed drafts of the paper, and approved the final draft.

## Animal Ethics

The following information was supplied relating to ethical approvals (i.e., approving body and any reference numbers):

William & Mary Institutional Animal Care and Use Committee provided full approval for this research (IACUC-2019-09-22-13861-jpswad).

## Data Deposition

The raw data and script are available in the Supplementary File.

## Supplemental Information

Supplemental information for this article can be found online at http://dx.doi.org/10.7717/peerj.13313#supplemental-information.

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
