# Peer review of "Evaluating acoustic signals to reduce avian collision risk"

_PeerJ, doi:10.7717/peerj.13313_

## Round 0.1 · original submission · Major Revisions

We have received three reviews for your manuscript. In light of the comments received, a number of major revisions are needed on your work, especially concerning points raised by reviewer one.

More specifically, in your revision you should pay a particular attention to 1) rethink the general context/framework of the paper, 2) adjust the statistical reporting of results (as well as R scripts and data provided) and 3) clarify the interpretation of results and consider other potential confounding factors when drawing conclusions. All other comments provided also need to be addressed convincingly.

Reviewer 1 ·

Basic reporting

1. Questionable relevance of experimental context for window and light-mediated collisions

The authors frame their paper in the context of declines in bird populations due to people, stating that collisions make up a meaningful part of that mortality. I don't disagree; collisions lead to hundreds of millions of deaths annually in the US along (e.g. Loss et al. 2015). However, the vast, vast majority of those collisions occur because of glass, light, or a combination of the two. Unfortunately, the results of this study don't seem applicable to these scenarios--as the authors state in the discussion, "sound is unlikely to reduce (or may even increase) the risk of a collision" with windows (lines 376-377) or when artificial light is involved (line 381). And the experimental set up (a looming black object in a flight chamber) is very different than, e.g., a window.

Relatively speaking, a tiny amount of collisions occur because of birds striking a solid, non reflective object. For example, Loss et al. 2015 estimate that there are 234,000 annual collisions with wind turbines in the US, compared to 599 million collisions with building windows. Mitigating these collisions is of course a worthwhile endeavor, but I find it disingenuous to frame the study in the context of the "hundreds of millions of bird deaths" (line 43) when sound doesn't have relevance for--or may even worsen--the vast majority of those events.

Experimental design

The experiment seems well designed.

Validity of the findings

2. Statistical reporting

I found it odd that there were no figures or tables corresponding to the main results of treatment vs control flights (described in the text on lines 262-268). Please provide both figures and tabular results for these comparisons.

Especially in an experimental context, reporting estimated effect sizes (and confidence intervals) of the results is important. In the main text, the authors report F- and p-values but not effect size estimates and confidence intervals. I would ask the authors to please report effect estimates in every instance where F- and p-values are reported, and to provide confidence intervals in tabular form (e.g. in Table 3).

I reviewed the provided R script and data.
- The authors provide summarized measures for each flight (those given in Table 2), but not the original instantaneous data from which they derived these summaries. The data given is enough to reproduce the statistics, but not the calculation of those original metrics. Therefore, I am not sure if "all underlying data have been provided" as required by PeerJ.
- The statistical models from which the authors calculated whether the frequency and shape of the sound have an effect are mixed-effects models with bird ID as a random effect. This is not reported anywhere in the main text. The authors should fully describe their modeling approach.
- I was not able to run the code for any of the anova_test functions - all of the tests gave me errors.
- I did not see code corresponding to the main results of treatment vs control flights (described in the text on lines 262-268), but I may have missed it because of the errors.

3. Are birds just avoiding a loud noise?

From the results of the experiment, it is difficult to disentangle whether the behaviors the authors report when sound was played were due to the birds recognizing the obstacle earlier in their flight, or whether they were just avoiding the loud noise. The latter scenario would still be useful, but the authors frame their study largely with the former. Indeed, in lines 239-241, the authors state, "assessing distance from the speaker and from the obstacle separately allowed
insight into whether avoidance behaviors were in response to the warning signal or visual detection of the flight hazard." In the results, they report statistically significant effects on both measures (lines 264-268). The larger F-value for the distance from the speaker (28.6, vs 21.4 for the obstacle) suggests that birds may have more consistently avoided the speaker, although it seems like the average effect was likely very similar. So it seems essentially impossible to distinguish these two scenarios. Therefore, I am unconvinced when on lines 323-325 the authors conclude that the birds were "aware of the obstacle."

I also wonder if measuring distance from speaker is of limited use if "the speaker emitted a highly directional sound field that was approximately constant in amplitude at all distances from the speaker" (lines 162-163). If the sound was equally loud at all distances, how could the bird judge where the sound was coming from?

Additional comments

Line 79 - what is the evidence that insufficient visual perception is why those methods (decals, lights, lasers) don't work as well?

In the introduction, the authors talk generally about "birds" when discussing eye and skull morphology. I would prefer that they specify the groups of birds to which they refer (e.g. songbirds), as many birds do have excellent binocular vision (e.g. birds of prey).

Line 219: should be "for EACH FRAME of the flight" - it currently sounds like these metrics were calculated only once per flight.

Fig 4 - I like that all the data points are shown in addition to the error bars (although they are hard to see - suggest making the points a little lighter and the error bars bolder).

Reviewer 2 ·

Basic reporting

The is a well written and presented ms. It meets all of the criteria for basic reporting.

Experimental design

The ms. fulfills all of the requirements of good experimental design as defined by the journal. The research question is well defined and placed in a clear context and the analysis is rigorous. However, I do have some concerns which are described in section 4.

Validity of the findings

This ms. meets all of the criteria

Additional comments

I have a concern about what the results actually show. Certainly in the presence of an obstacle the presentation of a sound does result in a change in the birds' flight paths. However, is this necessarily an effect that occurs only in the presence of an obstacle? What is the basic mechanism that is operating? The authors have not shown that the sound results in the birds actually avoiding the obstacle, they have simply shown that their flight behaviour changes when a sound is present but this could actually enhance the probability of collision.

The assumption seems to be that projecting the sound at the bird increases its awareness of the obstacle. This may work by making a bird more visually aware of its surroundings, and in consequence the bird takes more note of what lies ahead. However, the sound may simply be changing a bird's flight behaviour through avoidance of the sound. A further control condition, in which the tarp obstacle is absent, may sort out whether the birds are seeking to avoid the sound rather than the sound acting to alert the bird to detect an obstacle using visual cues. If a bird behaves the same when sound is given regardless of the presence of the obstacle then this would suggest that its effect is not specifically to do with collision avoidance.

A related problem arises in the use of the term "warning signal", as in lines 99 and 113, but also elsewhere. To be a "warning" signal the bird needs to learn a pairing of the sound with the obstacle. The experimenter assumes that a sound will warn the bird, but what can it warn the bird of? The association between a sound and a sheet of tarpaulin is unlikely to be innate, such an association would have to be learned. In humans, warning signals are culturally determined, we learn what a warning signal is, we learn that it is a sound that carries specific information. If we have not learnt the association then an un-predicted acoustic signal may simply alert/arouse our attention, but it does not warn us. I would suggest the authors consider rewriting some of the ms. so as not to use the idea of warning. At root, this is getting at the difference between the sensory salience of a signal (that is the features that are most likely to be detected; e.g. manipulation of frequency, sound shape, volume) and its cognitive salience, i.e. what does the bird interpret that signal to be?. Does it actually warn of something specific or does it have a non-specific arousal function?

·

Basic reporting

It is an interesting manuscript on collision avoidance in birds. A not so common experimental approach which must be acknowledgment

Experimental design

I did not find any shortcoming in general. They provide a good enough description
the only problem for my is that they did not conducted any experiment on habituation,

The potential individual variability in responses must be adressed

Validity of the findings

I have some doubt about habituation and the effect of individulas varience in behaviour

---

## Round 0.2 · Minor Revisions

Like reviewer 1, I am pleased with the revisions performed on the manuscript. There are only a few additional comments listed below that need to be integrated.

Reviewer 1 ·

Basic reporting

Please mention in the abstract that the results of this study most applicable to collision scenarios with solid non reflective objects, and not for window/glass or when artificial light is involved.

Regarding the R script:

(1) All the code ran successfully on my machine.

(2) The data paths are given as local paths on Robin Thady’s computer. For example:

deltaV <- read.csv("/Users/robinthady/Desktop/Thady manuscript/Raw Data/deltaV.csv")

I would suggest rewriting the code so only relative paths are used, which will make it easier for a user to run the script on their own machine after specifying a working directory.

Regarding the raw data provided:

I had a look at the “Bird_tracking” data folder. I thank the authors for including the raw data. However, I found the data difficult to decipher. I could not find a readme or any other document describing how the data are structured and what each directory and file represents. Please provide this information, otherwise having provided the data will ultimately of limited use. It would be helpful to provide a script showing how the flight data can be processed to obtain the summaries that went into the main analysis.

Experimental design

The paper satisfies all primary requirements.

Validity of the findings

The paper satisfies all primary requirements.

Additional comments

I think the authors for a thorough and well done revision.

---

## Round 0.3 · accepted · Accept

I am satisfied by the final revisions made on the manuscript.